# Ibero-American Society of Interventionism (SIDI) and the Spanish Society of Vascular and Interventional Radiology (SERVEI) Standard of Practice (SOP) for the Management of Inferior Vena Cava Filters in the Treatment of Acute Venous Thromboembolism

**DOI:** 10.3390/jcm11010077

**Published:** 2021-12-24

**Authors:** Miguel A. De Gregorio, Jose A. Guirola, Sergio Sierre, Jose Urbano, Juan Jose Ciampi-Dopazo, Jose M. Abadal, Juan Pulido, Eduardo Eyheremendy, Elena Lonjedo, Guadalupe Guerrero, Carolina Serrano-Casorran, Pedro Pardo, Micaela Arrieta, Jose Rodriguez-Gomez, Cristina Bonastre, George Behrens, Carlos Lanciego, Hector Ferral, Mariano Magallanes, Santiago Mendez, Mercedes Perez, Jimena Gonzalez-Nieto, William T. Kuo, David Jimenez

**Affiliations:** 1Group GITMI, Minimally Invasive Image Guided Surgery, Interventional Radiology Department, Universidad de Zaragoza, 50009 Zaragoza, Spain; mgregori@unizar.es (M.A.D.G.); carolse@unizar.es (C.S.-C.); joserodriguez.vet@gmail.com (J.R.-G.); cbonastr@unizar.es (C.B.); 2Department of Hemodynamics and Endovascular Therapeutics, Hospital Universitario Austral, Buenos Aires 1500, Argentina; sergio.sierre@usa.net; 3Department of Interventional Radiology, Hospital Ramón y Cajal, 28034 Madrid, Spain; jurbano34@gmail.com; 4Department of Interventional Radiology, Hospital Virgen de las Nieves, 18014 Granada, Spain; juanciampi@hotmail.com (J.J.C.-D.); pedropardomoreno@gmail.com (P.P.); 5Department of Interventional Radiology, Severo Ochoa, 28914 Madrid, Spain; jmabadal@yahoo.es; 6Department of Interventional Radiology, Hospital Universitario Dr. Negrín, 35010 Las Palmas de Gran Canaria, Spain; juanpulidoduque@yahoo.com; 7Department of Interventional Radiology, HC Alemán Buenos Aires, Buenos Aires 1640, Argentina; eeyheremendy@hospitalaleman.com; 8Department of Interventional Radiology, Hospital Universitario Dr. Peset, 46017 Valencia, Spain; elonjedo@gmail.com; 9Department of Interventional Radiology, Hospital General de México, Ciudad de Mexico 06720, Mexico; gpeguerrero57@gmail.com; 10Department of Interventional Radiology, HU de Cartagena de Indias, Cartagena de Indias 130000, Colombia; micaelarrieta@gmail.com; 11Department of Interventional Radiology, VIR Chicago at Hinsdale Adventist Hospital, Hinsdale, IL 60521, USA; georgebe@gmail.com; 12Department of Interventional Radiology, Complejo Hospitalario de Toledo, 45007 Toledo, Spain; clanciego@gmail.com; 13Department of Interventional Radiology, Louisiana State University, New Orleans, LA 70112, USA; hectorferral@gmail.com; 14Department of Interventional Radiology, Hospital Povisa, 36211 Vigo, Spain; presidente@servei.org; 15Department of Interventional Radiology, Hospital Puerta de Hierro, 28222 Madrid, Spain; smendez.sma@gmail.com; 16Department of Interventional Radiology, Hospital Vall d’Hebrón, 08035 Barcelona, Spain; merchebrea@yahoo.es; 17Department of Interventional Radiology, Hospital Clínico San Carlos, 28040 Madrid, Spain; jimenagn@gmail.com; 18Division of Vascular and Interventional Radiology, Stanford University Medical Center, Stanford, CA 94305, USA; wkuo@stanford.edu; 19Pulmonology Department, Hospital Universitario Ramón y Cajal (IRYCIS), 28034 Madrid, Spain; djimenez.hrc@gmail.com

**Keywords:** vena cava filters, retrievable, clinical practice guideline on venous thromboembolic disease

## Abstract

Objectives: to present an interventional radiology standard of practice on the use of inferior vena cava filters (IVCFs) in patients with or at risk to develop venous thromboembolism (VTE) from the Iberoamerican Interventional Society (SIDI) and Spanish Vascular and Interventional Radiology Society (SERVEI). Methods: a group of twenty-two interventional radiologist experts, from the SIDI and SERVEI societies, attended online meetings to develop a current clinical practice guideline on the proper indication for the placement and retrieval of IVCFs. A broad review was undertaken to determine the participation of interventional radiologists in the current guidelines and a consensus on inferior vena cava filters. Twenty-two experts from both societies worked on a common draft and received a questionnaire where they had to assess, for IVCF placement, the absolute, relative, and prophylactic indications. The experts voted on the different indications and reasoned their decision. Results: a total of two-hundred-thirty-three articles were reviewed. Interventional radiologists participated in the development of just two of the eight guidelines. The threshold for inclusion was 100% agreement. Three absolute and four relative indications for the IVCF placement were identified. No indications for the prophylactic filter placement reached the threshold. Conclusion: interventional radiologists are highly involved in the management of IVCFs but have limited participation in the development of multidisciplinary clinical practice guidelines.

## 1. Introduction

Inferior vena cava filters (IVCFs) are metallic devices designed specially to prevent venous thrombi larger than 3 mm from migrating from the lower extremities to the lung [1]. There appear to be no questions about the effectiveness of IVCFs even though there is no significant evidence [2]. However, the placement of an IVCF has always been controversial since there is no clinical evidence to support its efficacy. There are some unquestionable indications for IVCFs, universally accepted by the scientific community for the prevention of venous thromboembolism (VTE), and these are contraindications to anticoagulant therapy, a hemorrhagic complication of anticoagulant therapy (ACT), and the recurrence of VTE despite correct anticoagulation [3,4]. The indication of an IVCF in recurrent pulmonary embolism (PE) is discussed in some guidelines since it would be necessary to review “the correct anticoagulation therapy” [4,5,6]. However, in daily practice, IVCFs are used in other clinical indications, such as prophylaxis for VTE high-risk patients, polytraumatized patients, bariatric surgery, and for patients with life-threatening massive pulmonary embolism (PE), that require surgical thrombectomy or catheter-directed treatment (CDT) (thrombectomy and local mechanical/pharmacological thrombolysis). On the other hand, there are strict indications established by scientific societies and more accepted indications in daily medical practice [7].

Perhaps the most ambitious project concerning IVCFs is the multicenter prospective study PRESERVE (NCT02381509) that involves a total of 60 centers with more than 1800 IVCFs placed. Their results may provide substantial information on the safety and efficacy of these devices [8]. The clinical guidelines between scientific societies and medical specialties are very important for the knowledge diffusion regarding IVCFs, the indications for placement, the proper management of patients with IVCFs, and advising the compliance with the current clinical guidelines.

For more than 70% of the patients, interventional radiologists (IRs) are responsible for the IVCF placement, care, and recovery in most countries [9]. However, it is striking that the vast majority of the scientific and consensus guidelines on venous thromboembolic disease do not involve IRs in their preparation [4,5,6].

The responsible physicians for the placement and retrieval of more than 70% of IVCFs in most countries are interventional radiologists (IRs) [9]. However, it is striking that the vast majority of the scientific and consensus guidelines on venous thromboembolic disease do not involve interventional radiologists in their preparation [4,5,6].

In the present clinical guideline, the authors from the Ibero-American Interventional Societies (SIDI) and the Spanish Interventional and Vascular Radiology Societies (SERVEI) provide consensus indications based on evidence regarding recommendations for the placement, retrieval, and management of IVCFs. In addition, it is appropriate that IRs contribute to the dissemination among their partners of the correct management and use of these controversial devices.

## 2. Material and Methods

A group of interventional radiologist experts were united online from the Ibero-American Interventional Society (SIDI) and the Spanish Interventional and Vascular Radiology Society (SERVEI) to determine the current guidelines and available evidence for the management of IVCFs, to establish if IRs participate in the development of these current scientific guidelines, and to develop an interventional radiology standard of practice for the management of IVCFs. Ethics approval was not required for this clinical practice guideline because it did not involve human participants.

The coordinator of the panel was nominated by the scientific societies (SERVEI and SIDI) and subsequently nominated panelists, ensuring proper expertise in the subject, and they were also reviewed for potential COIs. The selection of experts was made on the basis of the existing pulmonary thromboembolism working groups of both societies. Many of the experts had already participated in IVCF projects. Several meetings of the group of experts were held, which were online meetings. All these meetings were coordinated by the principal investigator of the expert group. The final panel consisted of twelve IRs from SERVEI, ten Drs from SIDI, and two external consultant experts in the treatment of VTE and management of IVCF (one interventional radiologist from the Society of Interventional Radiology (SIR), and one pulmonologist from Spanish Society of Pulmonology and Thoracic Surgery (SEPAR)).

The subcoordinators of the project (J.A.G. and J.J.C.D.) conducted a comprehensive search of MEDLINE/PubMed. The literature search was approved by the panelists to determine the participation of interventional radiologists in current clinical guidelines in the management of IVCFs. The search was performed 20 October 2020 using Boolean operators ‘AND’ and ‘OR’, and the final result in the PubMed Advanced Search Builder was: ((((((guideline) OR (expert consensus)) OR (expert panel)) OR (society guidelines)) OR (clinical guidelines)) OR (study group)) AND (((((filter vena cava) OR (vena cava filter)) OR (interruption of vena cava)) OR (inferior vena cava filter)) OR (interruption device inferior vena cava)). All available manuscripts from 1991 to the present date were evaluated and limited for significant evidence-based publications in either Spanish or English language. 

Questions were developed in three categories for the indication of placement of IVCF: *absolute, relative, and prophylactic indications for the placement of any type of IVCF versus not placing IVCF*, the utilization of retrievable filters vs. permanent filters, and the retrieval of IVCF (Abbreviations). Each question was answered by each panelist with three options: 1. agree with the current indication, 2. should be considered, 3. not agree with the indication. Voting was determined as a poor agreement when 0–33% of the agreement was achieved, moderate agreement when 33–66% was achieved, and high agreement when >66% was achieved.

The experts from both societies received a questionnaire via email and worked on a common draft. The manuscript was revised after panelist consideration and feedback, and the final draft was sent to the two expert consultants for final considerations.

## 3. Results

### 3.1. Interventional Radiologist Participation in the Development of Multidisciplinary Clinical Guidelines in the Management of Inferior Vena Cava Filters

In most countries, interventional radiologists are responsible not only for the placement but also for the retrieval of the IVCF as well as performing catheter-directed thrombolysis and/or pharmacological or mechanical thrombectomy-CDT for the treatment of PE. However, in massive and symptomatic proximal deep vein thrombosis (DVT), they have not traditionally been invited to participate in the development of consensus guidelines for the management of VTE.

A total of 233 articles were obtained, from which titles and abstracts were reviewed. Of them, 79.39% (185 articles) were excluded as non-significant evidence-based scientific publications (comment, editorial, letter, news, newspaper, or case report), or duplicates. A total of forty-eight articles (20.6%) were included, which were reviewed in their entirety. Only eight (3.4%) articles corresponded to clinical guidelines of societies or associations [4,5,6,10,11,12]. In no clinical guidelines did interventional radiologists participate, but the following were included: internists, cardiologists, hematologists, general surgeons, traumatologists, vascular surgeons, pulmonologists, angiologists, and thoracic surgeons. However, only two clinical guidelines corresponded to radiological or interventional radiology societies: Society of Interventional Radiology—SIR [13] and American College of Radiologists—ACR [14], both in which diagnostic and interventional radiologists participated. 

### 3.2. Compliance with the Current Clinical Guidelines

Despite randomized trials [4,15] and multiple recommendations in successive guidelines, various studies [10,11,16,17] have demonstrated no reduction in mortality in patients with IVCF and with DVT. More than 36,000 IVCF were placed in the USA in patients with DVT [18]. In the study by Spencer FA et al. endorsed by the NIH (National Institute of Health), the placement of an IVCF according to the guidelines was only in 51% of the patients studied with VTE [19]. Baadh AS et al., in a study conducted in a single center on 957 IVCFs placed by different specialties, found that the ratio of adherence to the ACCP/SIR guidelines was 44% for IRs, 40% for vascular surgeons, and 34% for cardiologists [20]. Despite everything, the true proportion of adherence to the current clinical guidelines regarding the placement of IVCFs is unknown. Possible causes for the poor adherence to the recommendations could be new retrievable filters with additional clinical indications [21], insufficient diffusion of the current published clinical guidelines, the different indications and lack of agreement between scientific societies, and scarce participation of the different specialists who are directly involved in the management and placement of the IVCF [22].

### 3.3. Venous Flow Interruption in IVC and Types of Filters

IVCFs can be permanent, retrievable, bioabsorbable, and bioconvertible [22,23,24]. Permanent filters with more than 40 years of experience are devices made of different alloy metals, mainly stainless steel and nitinol; they have a filtering capacity of embolized thrombus from lower limbs, and their clinical efficacy has been demonstrated in numerous studies [2,25,26,27].

Retrievable filters are very similar in morphology and design compared to permanent filters, but they are equipped with a small hook that allows them to be caught by special recovery devices, such as lasso or snare catheters, or even with surgical forceps that help the extraction without great difficulty from the inferior vena cava (IVC) [28,29,30] (Figure 1).

Several studies regarding permanent filters with a follow-up of 6 to 18 months have shown high efficiency in the filtration of these devices, observing pulmonary embolism (PE) between 2.6 and 3.8%; however, IVCFs also produced between 6 and 32% DVT and between 3.6 and 11.2% IVC thrombosis [26,31]. The efficacy in the prevention of PE of retrievable IVCFs is similar compared to permanent IVCFs, with PE rates between 2 and 4%, with a DVT of 5–15% and an IVC thrombosis of 0.6–8% [22,30,32,33] (Box 1).

There are also special devices, such as the Angel^®^ Catheter (Bio2 Medical), which consists of a filter permanently attached to a three-lumen venous catheter. It protects against PE in patients at a high risk for VTE without contraindication to anticoagulation for short periods (4–8 days) without having documented serious complications [34,35,36]. It offers the advantage of being able to be placed at the bedside without angiographic control, although the possibility of fractures of the catheter for more than 30 days of indwelling time of the filter has been reported [37,38].

Box 1SIDI & SERVEI Recommendation.
**Types of Filters**
The benefit of retrievable filters is demonstrated in the literature [13,30,39]. However, there is no significant evidence regarding the placement of retrievable versus permanent since there have been no randomized studies. The placement of retrievable filters instead of permanent filters is recommended because of the ability of retrieval when filtration is no longer needed. 

### 3.4. Safety and Efficacy of IVC Filters for Acute VTE

In 2015, Mismetti P et al., in their randomized study of 400 patients with PE and DVT, compared two groups: the first group consisted of patients with ACT and IVCF (200 participants), and the other group just ACT (199 participants). No significant differences were observed in recurrent PE at 3 and 6 months [15].

The study performed by Decousus et al. [2] has been the scientific basis on which they have relied heavily on the consensus of medical societies to restrict the use of IVCF. However, despite being a very solid study, immediately after its publication, some authors with great expertise in the management of IVCF answered with comments indicating that the study by Decousus et al. was a segregated study with a small population with important methodological flaws, generalizing the findings of four types of permanent filters already withdrawn from the market to all other types of available retrievable filters [18,40,41].

Recently, a Cochrane systematic review concluded, of the six identified randomized studies, one did not show significant evidence in the placement of an IVCF in the first three months regarding PE, DVT, severe bleeding, and mortality, and another study also showed no benefit in PE and mortality for patients with prophylactic IVCF placement in polytraumatized patients. The remaining four studies presented no firm conclusions regarding filter efficacy in the prevention of PE, suggesting the necessity of further trials to determine the efficacy of IVCFs [42].

The American College of Chest Physicians (ACCP), in its recommendations since the seventh conference on antithrombotic and thrombolytic therapy [16] and successive conferences to date, advises against the use of an IVCF outside the proper indications [3].

The multiple retrospective studies in the medical literature regarding IVCFs have not demonstrated clear clinical evidence regarding their efficiency, and the few prospective randomized studies have not only failed to demonstrate efficacy but have, on the contrary, shown a lack of benefits in the use of an IVCF and a notable increase in the rate of complications [2,15,25,26,27,28].

Inferior vena cava filters can present complications in their management, but, according to the FDA (Food and Drug Administration) MAUDE office (Manufacturer and User Facility Devices), in 2014, it showed that the vast majority of reported complications were related to retrievable IVCFs (86%) compared to 13.2% of permanent filters. In this communication, the most frequent complications were those related to the procedure of filter insertion (45%), of which 27.1% were penetration of the filter legs into the IVC wall >2 mm [35,36]. The FDA itself advises a correct and obvious indication of the IVCF and to schedule its withdrawal between 29 and 54 days [43].

A recent review by Jia Z et al. including 9002 patients with 15 different types of IVCFs (permanent and retrieval) showed 19% of all IVCFs (1699/9002) had penetration (>3 mm) into the wall of the IVC. In a total of 19% of patients with IVC penetration (322/1699), there was evidence of adjacent structures or organ involvement. The most frequent symptom was pain (8%), and the most important complication was abdominal bleeding (2%). The major complication with penetration of the IVC wall (0.5%—83 patients) required intervention, including open surgery [44].

However, some investigators from the RIETE group (Computerized Registry of Patients with Venous Thromboembolism) observed a 65% reduction in mortality after the placement of an IVCF in a cohort of patients with PE with high bleeding risk compared to another cohort with similar characteristics without IVCFs [45].

### 3.5. Indications and Contraindications Placement of IVCF

Most of the retrievable IVCFs can be used, when necessary, as permanent filters; therefore, the indications and contraindications refer to both types of filters. Classically, absolute, relative, and prophylactic indications have been accepted (Table 1).

The PREPIC2 study shows that there were no significant differences in recurrence in both groups with single anticoagulation or IVCF anticoagulation at three and six months [15].

#### 3.5.1. Absolute Indication for Placement of Inferior Vena Cava Filters

The absolute indications are accepted and supported by sufficient medical literature. Most medical societies accept them [3,4,5,10,11,13]. Although some authors have questioned the efficacy of filters in patients with ACT and recurrent PE [15], Mellado et al. in their study demonstrated a significant decrease in the recurrent PE group treated with IVCFs [46].

There are limited data regarding patients with recurrent PE who develop chronic PE and pulmonary arterial hypertension. There is no clear positioning, although an IVC filter is occasionally implanted with long-term ACT [47].

The update of the SIR clinical guideline in 2020 [13] does not clarify the position of this medical society regarding the relative indications. On the other hand, the medical literature shows controversial arguments in both senses regarding the placement of IVCFs in these relative indications [39] (Box 2).

Box 2SIDI & SERVEI Recommendation.
**IVCF Absolute indications for filter placement in acute VTE with PE**
The placement of IVCF is recommended in the case of VTED with PE.(a) Contraindication for anticoagulation (100% agreed 22/22)(b) Bleeding complication due to anticoagulation (100% agreed 22/22)(c) Recurrent PE despite correct anticoagulation * (86% agreed 19/22)★★★ 100% of the panelists agreed to place IVCF (22/22) for a) or b).* It is advisable to consider new ACT or adjust correct anticoagulation, if current anticoagulation has failed, before IVCF placement.

#### 3.5.2. Relative Indication for the Placement of Inferior Vena Cava Filters

Regarding the relative indications, there is no clinical evidence for the placement of IVCFs, and they are based on the experience of experts; however, some scientific societies have accepted them as indications. The SIR (American Society of Interventional Radiology), in its previous guidelines [17], accepted relative and prophylactic indications, such as filter before fibrinolysis, thrombus of the iliocaval region, and trauma with high VTE risk, but, in the current update of its latest guideline, it no longer recommends them as indications for the placement of an IVCF [13] (Table 2) (Box 3).

Box 3SIDI & SERVEI Recommendation.
**IVCF placement for relative indications**
The placement of IVCFs is not recommended as a routine for relative indications. In patients with DVT and poor respiratory reserve or preexisting cardiorespiratory disease, ACT should be considered before the use of IVCFs [3,4,13]. ★★★ 100% of the panelists agreed to consider and individualize based on the particular characteristics of each patient for the placement of IVCF.

##### Filter Placement in Hemodynamically Unstable Patients with VTE

In massive or high-risk PE, as well as in patients with poor pulmonary reserve, some societies recommend considering the use of IVCFs. Two studies conclude that there is no benefit of an IVCF compared to ACT [15,27]. Both studies were conducted in patients with PE or DVT but not in hemodynamically unstable patients. In the comparative study by Billett HH et al. [27], the IVCFs were not withdrawn, while, in the series by Mismetti P et al., 76.5% of the IVCFs were withdrawn at three months [15]. For these reasons, the ACCP (American College of Chest Physicians) and ESC (European Society of Cardiology) do not recommend the use of IVCFs in patients with PE and/or DVT, only if there is a contraindication or complication with the ACT [3,4]. The SIR considers the option of an IVCF in selected patients with massive PE who undergo local thrombectomy and/or fibrinolysis [13]. Other studies with systematic use of IVCFs in high-risk PE have had one-month survival of 90%, and a success retrieval rate for IVCFs of 100% [48] (Box 4).

Box 4SIDI & SERVEI Recommendation.
**IVCF placement High-risk PE with CDT**
The indiscriminate use of IVCFs is not recommended in patients with PE and hemodynamic alteration with or without CDT [3,4,15]. However, the latest SIR recommendations for patients with massive PE and the use of CDT techniques recommend assessing the possibility of IVCF [13]. ★★☆ 59% of the panelists agreed to place IVCF (13/22).

##### Filter Placement in Recurrent PE or Chronic PE

There are limited data regarding patients with recurrent PE or chronic PE who develop pulmonary arterial hypertension. There is no clear positioning, although IVCFs are occasionally placed with long-term ACT [47] (Box 5).

Box 5SIDI & SERVEI Recommendation.
**Filter placement in pulmonary hypertension and chronic PE**
The update of the SIR guideline in 2020 does not clarify the position of this medical society regarding the relative indications [13]. On the other hand, the medical literature shows controversial arguments in both senses regarding the implantation of IVCFs in these relative indications [39].★★★ 100% of the panelists agreed not to place IVCF (22/22).

##### Placement of IVCF before CDT in Proximal DVT

There is controversy, but there is scientific evidence on the use of an IVCF in selected patients with proximal DVT who will undergo fibrinolysis and mechanical thrombectomy [49,50]. Lee SH et al., in their study of 70 patients with CDT for proximal lower-extremity DVT, suggest that the placement of retrievable filters during the procedure is effective for protecting against PE or lethal complications [51]. Others, however, do not find them useful during the CDT of the lower extremities since, in their studies, they observed very few migrated thrombi [52] (Box 6).

Box 6SIDI & SERVEI Recommendation.
**Placement of IVCF before CDT in proximal DVT**
In selected cases in which it is necessary, an IVCF could be placed before the procedure, and it is necessary to retrieve it once the procedure is finished, or a retrievable IVCF should be used in centers where there is extensive experience with implantation and retrieval of filters and direct patient follow-up [49,50].The indication for placement of IVCF before CDT (thrombolysis or catheter thrombectomy) in proximal DVT received 77% agreement. In the case of using a filter, use it temporarily and remove it as soon as possible or immediately after performing CDT.★★★ 77% of the panelists agreed to place IVCF (17/22).

##### Prophylactic Indication for the Placement of Inferior Vena Cava Filters

The prophylaxis of PE by the placement of an IVCF in high-risk patients without proven DVT and with a temporary contraindication to anticoagulation has increased the placement of IVCFs in the past, especially with the appearance of retrievable filters [39]. There are no well-defined data or clinical evidence to support this indication (Box 7). 

Box 7SIDI & SERVEI Recommendation.
**Prophylactic indication for the placement of inferior vena cava filters**
The ACCP, ESC, and AHA (American Heart Association) do not recommend the use of IVCF in the absence of DVT [3,4,5]. Although the position of the SIR was somewhat different in the past [17], the current guidelines recommend against the use of IVCF in patients with severe trauma, major surgery, bariatric surgery, and long-immobilized patients without DVT [13]. However, each patient must be individualized, assessing risks and therapeutic possibilities.It is not recommended to use IVCF for prophylaxis indications. Only 13% (3/22) voted in favor of the indication: trauma with a high risk of VTE without the possibility of ACT [35].Unanimity was reached (100%) in the non-use of IVCF for prophylaxis purposes.★★★ 100% of the panelists agreed not to place IVCF as a prophylaxis indication (22/22).

## 4. Contraindications

Contraindications to IVCF placement are rare but easily recognizable. Two types of contraindications are considered: absolute include a total thrombosis of the IVC, inferior vena cava with a diameter greater than 40 mm and non-correctable coagulopathy, while the relative contraindication would be sepsis and partial thrombosis of the IVC that prevents the placement of the filter [53,54].

## 5. Removal of Retrievable Inferior Vena Cava Filters

The IVCFs were designed to stop thrombi on their way to the pulmonary territory and to be recovered after their need or to remain in the IVC for life. To do this, the IVCF had to have a less secure clamping and fixation system to the vena cava that allows easy recovery [30,39]. However, several studies have shown that the removal of a temporary IVCF can also cause complications and that these complications are generally more likely the longer the indwelling time of the filter [25,55,56]. Since 2010, the FDA has issued warnings of the danger of IVCFs and more recently of the need to remove them as soon as possible between 29 and 54 days [13,43].

However, the reality in daily practice is that fewer IVCFs are removed than initially programmed. The recovery rate is very low (20–30%) of the total filters placed [9,22,32,33]. The main reason does not lie in technical difficulty but in timing and indication [48,57,58,59]. 

Who is responsible for the withdrawal of the IVCF, the physician who indicates the placement of an IVCF or the interventional radiologist who places the IVCF in the patient? The responsible party for the IVCF retrieval should be the physician in charge of the patient’s follow-up and programmed proper retrieval. Moreover, a multidisciplinary working group should be implemented in which all the participants are involved in the diagnostic and therapeutic process [60,61].

Thus, a clinical follow-up of the patient with an implanted IVCF should be established to plan its removal and detect potential complications. Irwing E et al. and Rottenstreich A et al., with the follow-up visit, were able to increase the percentage of recovery of IVCFs from 64% to 84% and 10 to 50%, respectively [62,63]. In this sense, the creation of registries in medical societies related to IVCFs can also increase the percentage of filters recovered [57].

Following the technical recommendations and the analysis of current scientific evidence, SIDI and SERVEI present in this document their endorsement regarding the implantation and withdrawal of the IVCF (Table 3).

The PREPIC2 study shows that there were no significant differences in recurrence in both groups with single anticoagulation or IVCF anticoagulation at three and six months [15].

As already indicated, the IVCF retrieval rate does not depend only on technical difficulties but other reasons, such as organization, convenience, and awareness, which seem to be influential [4,20,59]. If it seems logical that the filter implantation time constitutes a degree of difficulty, technical skills and experience can solve IVCF retrieval with special techniques.

There is an indication to remove an IVCF whenever it is no longer necessary from a clinical point of view and when there are complications (migration, penetration of more than 3 mm into the caval wall with or without symptoms). To anticipate complications and facilitate removal, before its retrieval, it could be advised to perform an abdominal CT with a low radiation dose [48] (Box 8).

Box 1SIDI & SERVEI Recommendation.
**Withdrawal of inferior vena cava filter**
Remove the IVCF when it is no longer clinically necessary as soon as possible.The FDA recommends withdrawing the IVCF between days 29 and 54 [8,43].★★★ 100% of IRs agreed to withdraw IVCF as soon as possible when IVCF is no longer needed (22/22).

## 6. Conclusions

IVCFs are devices that have been proven to be effective in preventing thrombus migration from the abdomen and lower extremities to the lung. Its filters with special characteristics (metallic foreign body, morphology, and possible endothelial damage) imply a certain degree of local thrombogenicity. Therefore, an IVCF, when possible, requires ACT. IVCF implantation must be carried out following the clinical guidelines endorsed by each center and by the scientific evidence. The FDA has warned of the potential danger in the use of IVCFs and recommends the recovery of an IVCF when it is no longer clinically necessary and does not represent a benefit for the patient. The monitoring of all IVCFs is mandatory to schedule their removal and detect possible complications. When complications are detected in an IVCF, it will be mandatory to attempt its recovery, always evaluating the risk/benefit of the type of intervention. It would be desirable to have new devices that are effective and avoid complications. This clinical guide has some limitations since the impossibility for interventional radiologists to have face-to-face meetings due to the COVID-19 pandemic has also constituted an additional difficulty.

## Figures and Tables

**Figure 1 jcm-11-00077-f001:**
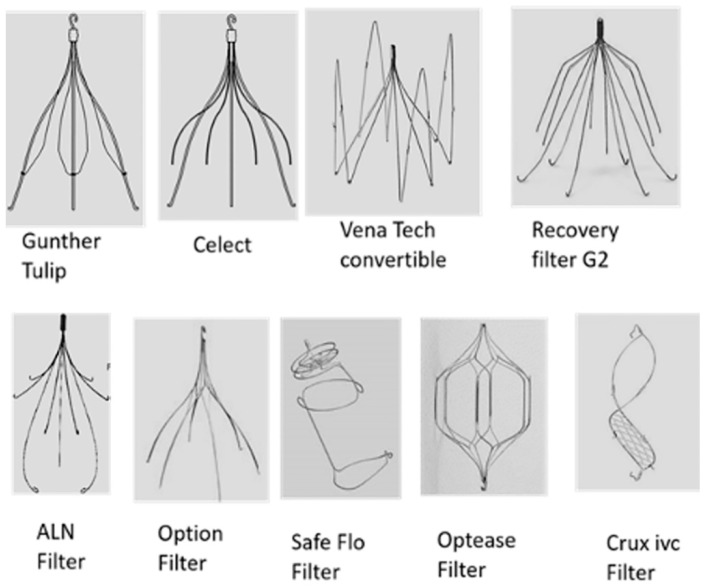
Different types of filters available for the treatment of venous thromboembolic disease.

**Table 1 jcm-11-00077-t001:** Indications of IVCF (modified from the SIR) [13,14,21].

Indications	Class	Level
Absolute		
Presence of VTE and contraindication for anticoagulation Recurrence of PE despite correct anticoagulation	IIa IIa	C C
**Relative**		
PE with poor lung reserve or right heart failure Chronic PE and pulmonary arterial hypertension Thromboendarterectomy Massive PE with thrombectomy or thrombolysis Thrombolysis of iliocaval thrombus Floating thrombus in the iliocaval sector	IIb IIb IIb IIb IIb IIb	C C C C C C
**Prophylaxis**		
Trauma patient with risk factors (immobility, large fractures, and inability to undergo anticoagulant therapy Paraplegia or other high-risk patients and inability to undergo anticoagulant therapy	IIb IIb	C C
**Routine**		
In all other situations, it is advisable not to implant IVCF	III	A

**Table 2 jcm-11-00077-t002:** Recommendations of the main clinical and radiology societies regarding IVCF [3,4,5,10,11,13]. The Implantation of temporary or retrievable Angel catheter-type IVCF should be performed in centers with extensive experience in IVCF extreme retrieval procedures and with an exhaustive follow-up.

Indication	ACCP	AHA	ESC	CIRSE	SIR	SEPAR
**Absolute**						
VTE with contraindication ACT	Yes	Yes	Yes	Yes	Yes	Yes
Major complication with ACT	Yes	Yes	Yes	Yes	Yes	Yes
PE recurrent despite correct ACT	No	Yes	Yes	Yes	Consider	Consider
**Relative**						
Massive PE and thrombectomy with/without fibrinolysis	No	Consider	No	Consider	Consider	-
Proximal floating thrombus	No	No	No	Consider	Consider	-
Thrombolysis/thrombectomy in proximal DVT	No	No	No	Consider	Consider	-
High-risk PE with poor pulmonary reserve	No	Consider	No	Yes	Yes	-
Chronic PE and PAH	No	No	No	Consider	No	-
Thromboendarterectomy	No	No	No	Consider	No	-
**Prophylaxis**						
Trauma with high risk without possibility of ACT	No	No	No	Yes	No	-
Paraplegia without the possibility of ACT	-	-	No	Yes	No	-
Surgery with high-risk VTE	No	-	No	Consider	No	-

Pulmonary embolism, DVT: deep venous thrombosis, PAH: pulmonary arterial hypertension, ACT: anticoagulation therapy, VTE: venous thromboembolism, ACC: American College of Chest physician, AHA: American Heart Association, ESC: European Society of Cardiology, CIRSE: Cardiovascular Interventional Radiology European Society, SIR: Society of Interventional Radiology, SEPAR: Spanish Society for Pulmonology and Thoracic Surgery.

**Table 3 jcm-11-00077-t003:** SIDI and SERVEI recommendations (modified SIR) [13,14].

Technical Recommendation	
Image Guidance	Fluoroscopic guidance is recommended for placement and retrieval of IVCF compared to other imaging techniques. However, preprocedural imaging for filter retrieval is not necessary; some authors recommend abdominal CT before retrieval of the filter to rule out complications [57,64,65].
Venous approach	Jugular, femoral, or brachial access can be used depending on operator skills, filter type, and favorable anatomy [13].
Duplicated inferior vena cava	There are two possibilities: placement of a filter in each vena cava or a single filter in a suprarenal localization [66].
Suprarenal placement of inferior vena cava filters	The main indication of the suprarenal filter is IVC thrombosis. Its retrieval should be the same as an infrarenal filter with a jugular approach [67].
Superior vena cava filter	Superior vena cava filters are not recommended because there are limited indications with a high-risk of filter migration. However, some authors admit the safety and efficacy of the procedure [68].
The positioning of the IVCF	It is important to ensure the position of the filter, avoiding angulations, tilting, or the introduction of the retriever hook into the renal veins. It is advisable to reposition it in the same procedure to guarantee proper placement of the filter and prevent future complications during retrieval [13].
Estimated retrieval time	There is no clear retrieval time. It is recommended when the IVCF is no longer necessary and should be removed as soon as possible. The FDA recommends between 29 and 54 days [55,69,70].
Anticoagulation therapy and inferior vena cava filters	If the patient has no contraindication for the ACT, it should be continued after filter placement, retrieval, or during the indwelling time of the IVCF. It is not necessary to discontinue the ACT for the retrieval procedure; however, it is advisable to determine a preoperative assessment of hemostasis before filter retrieval [57,71,72,73].
Hospitalization	Hospital admission is not necessary [13].
Recovery attempts	Sometimes, the IVCF, due to its positioning, inclination, or organ penetration, cannot be recovered on a first attempt. Additional maneuvers with special techniques are recommended. In the case of a lack of experience, it is recommended to send patients for advanced techniques for retrieval procedures in specialized centers. An IVCF can always be retrieved; however, it is important to evaluate the risk versus benefit of this procedure [13,74].
IVC filters in pregnancy	There are limited data regarding retrievable IVCF in patients with high-risk VTE in pregnancy. Complications seem to be higher in pregnant patients with thrombosis and IVC penetration [75].

## Data Availability

Not applicable.

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
