# Peer review of "Ibero-American Society of Interventionism (SIDI) and the Spanish Society of Vascular and Interventional Radiology (SERVEI) Standard of Practice (SOP) for the Management of Inferior Vena Cava Filters in the Treatment of Acute Venous Thromboembolism"

_jcm, 2021, doi:10.3390/jcm11010077_

Round 1

Reviewer 1 Report

The document presents the consensus of the relevant committees, providing Standards of Practice regarding IVCF. It is detailed enough but could be improved by language editing.

Author Response

Thank you

Reviewer 2 Report

The presented work is a standard operating procedure (SOP) to surgically place the inferior vena cava filter (IVCF). The SOP is intended for radiologists and is prepared by the Ibero-American Society of Interventionism (SIDI) and the Spanish Society of Vascular and Interventional Radiology (SERVEI). The authors give seven recommendations that include – use of retrievable IVCF, use during pre-existing conditions and anticoagulation treatment, its placement during different venous thromboembolism (VTE) scenarios such as different cases of pulmonary embolism (PE), high-risk PE with catheter-directed treatment (CDT), pulmonary hypertension, etc.

Major comments:

  1. The abstract, introduction and methods need to be rewritten clearly stating the role of radiologists in the placement of IVCF and the need for an SOP for them.
  2. The manuscript talks about reviewing 233 articles and many of them were discarded by the committee. These articles need to be referenced in the current paper for the readers to understand why these papers were discarded.
  3. Authors talk about absolute and relative indications – these need to be clearly specified to not confuse the reader.
  4. In recommendation 1 the authors talk about retrievable filters being better than permanent ones, but what about the bioabsorbable and bioconvertible ones. These different IVCFs were introduced by the authors themselves in the article.
  5. In table 1 there are terms such as IIa, IIb, and c. These terms need to be explained.
  6. In recommendation 4 what is meant by hemodynamically unstable patients.
  7. While placing the IVCF it is recommended to use Fluoroscopic guidance. Authors need to discuss different methods used at present and their advantages and disadvantages.
  8. The authors need to explain catheter-directed treatment (CDT) used in recommendation 5.
  9. Authors talk about IVCF removal as soon as possible when it is no longer necessary. Authors need to give examples of what happens when the IVCF is not removed on time. They also need to discuss the advanced and common IVCF retrieval process.

Minor comments:

  1. Line 106-109 the brackets seem to be wrong
  2. Line 116 - An appendix is referenced which is not present in the manuscript.
  3. Line 169 – Figure 1 is referenced which is not included in the manuscript.

Author Response

Reviewer 2

1.  The abstract, introduction and methods need to be rewritten clearly stating the role of radiologists in the placement of IVCF and the need for an SOP for them.The abstract, introduction and methods are modified following the instructions of the reviewers to emphasize the involvement of IRs in IVCFs and the need for SOPs for them

  1. The manuscript talks about reviewing 233 articles and many of them were discarded by the committee. These articles need to be referenced in the current paper for the readers to understand why these papers were discarded.

With the utmost respect for the reviewer, in our opinion of the 233 articles selected, only 48 were related to the subject. There is no place to cite an article that is not cited in the text. In the reviews, meta-analyzes, the excluded works are not usually included under certain criteria (see table I). However, if it is a trust problem, we attach a PDF (from the work excel) with the 233 articles in case you want to review them (Appendix II).

  1. Authors talk about absolute and relative indications – these need to be clearly specified to not confuse the reader

Thank you for your comment, the terms of: Absolute, relative and prophylactic indications are terms widely cited in the literature in relationship of filters usage. We believe that they are accurate.(Shah M et al Medicine 2008, Kaufman JA JVIR 2006) However, these would be the indications: absolute, relative and prophylactic

Absolute indication: Acute VTE while therapeutic on anticoagulation or in the presence of absolute contraindication to anticoagulation (recent neurosurgical procedure, major active or recent bleeding, coagulopathy).

Relative indication: Patient with acute or prior VTE considered at higher risk for either bleeding complications from anticoagulation or hemodynamic instability (transient bleeding, recurrent falls, multiple comorbidities, extensive PE, questionable compliance, central nervous system neoplasms, perioperative DVT, active cancer with potential for bleeding, poor cardiovascular reserve, ileocaval DVT, DVT with free floating thrombus).

Prophylactic: IVCFs were placed in the absence of current or prior VTE.

  1. In recommendation 1 the authors talk about retrievable filters being better than permanent ones, but what about the bioabsorbable and bioconvertible ones. These different IVCFs were introduced by the authors themselves in the article

There is not enough experience or publications to support the use of convertible or absorbable filters. However, we add this reference in recommendation 1

  1. In table 1 there are terms such as IIa, IIb, and c. These terms need to be explained.

The Class column classifies the levels of evidence in such a way that IIa is evidence based on non-randomized cohort studies and IIb from studies and generally multicenter case controls. The C in the Level column means that the evidence is controversial and does not allow recommendations to be made for or against

-Sackett Dl, Rosenberg WMC, Gary JAM, Haynes RB, Richardson WS. Evidence based medicine: what is it and what it isn´t. BMJ 1996;312:71-2.

West S, King V, Carey TS, Lohr KN, McKoy N, Sutton SF, et al. Systems to Rate the -Strength of Scienti€c Evidence. Health Services/Technology Assessment Text, National Library of Medicine. AHRQ Publication No. 02-E016, 2002. Disponible en:

https://www.ncbi.nlm.nih.gov/books/NBK33881/

  1. In recommendation 4 what is meant by hemodynamically unstable patients.

In general in the medical literature you can read unstable patients or hemodynamically unstable patients or  Hemodynamic instability means systolic blood pressure lower than 90 mmHg for more than 15 minutes or a drop of 40 mmHg of  BP or the need for inotropic drugs for maintenance of BP.

(Konstantinides  SV 2019, Kearon 2016)

  1. While placing the IVCF it is recommended to use Fluoroscopic guidance. Authors need to discuss different methods used at present and their advantages and disadvantages

Standard IVCF implantation should always be performed under fluoroscopic control, although in recent publication IVCF implantation with US or IVUS has been used with good results in special situations (LIU Y et al Ultrasound Med Biol 2015, Gunn AJ et al Vasc Endovascular Surg 2013)

US-guided techniques have been reported that mitigate transportation requirements in critically ill patients. Ultrasound guidance also obviates the need for ionizing radiation and nephrotoxic contrast administration.

 On the contrary, the biggest drawbacks are the lack of collaboration of the patient, obesity and anomalies of the vena cava

  1. The authors need to explain catheter-directed treatment (CDT) used in recommendation 5.

Thank you for your comments however we think that you were referring for the recommendation 4. In high-risk PE patients it is important to avoid a new embolism that worsens or could be fatal. On the other hand, instrumentation with catheters and instruments of different calibers could cause embolism

  1. Authors talk about IVCF removal as soon as possible when it is no longer necessary. Authors need to give examples of what happens when the IVCF is not removed on time. They also need to discuss the advanced and common IVCF retrieval process.

The FDA recommends removing the filter as soon as possible when it is no longer needed (29-54 days).

Filters are thrombogenic devices, so if anti-clotting is recommended when possible and, in addition, the fixation systems can damage the endothelium of the IVC, which leads to fibrosis and filter entrapment that makes it difficult to remove.

In general, the removal of IVCFs are simple procedures and they are removed without much difficulty, but there are circumstances such as significant inclination, inclusion of the removal hook in the wall of the IVC, filter embedded in the IVC, in these cases experience and availability of IVUS is required. That is why, specialized centers are required for patient referral for IVCF retrieval.

Minor comments:

  1. Line 106-109 the brackets seem to be wrong

Thank you, but if you put all the brackets in the pubmed search it will work

  1. Line 116 - An appendix is referenced which is not present in the manuscript.

The Appendix-I is uploaded again. Sorry for the inconvenience.

  1. Line 169 – Figure 1 is referenced which is not included in the manuscript.

The Figure-1 is uploaded. Sorry for the inconvenience.

Round 2

Reviewer 2 Report

The author's response to my previous comments is acceptable.

This manuscript is a resubmission of an earlier submission. The following is a list of the peer review reports and author responses from that submission.